# Evaluation of the Muscle Strength of the Tongue with the Tongue Digital Spoon (TDS) in Patients with Obstructive Sleep Apnea

**DOI:** 10.3390/life12111841

**Published:** 2022-11-10

**Authors:** Laura Rodríguez-Alcalá, Felipe Benjumea, Juan Carlos Casado-Morente, Peter M. Baptista, Carlos O’Connor-Reina, Guillermo Plaza

**Affiliations:** 1Otorhinolaryngology Department, Hospital Quironsalud Marbella, 29603 Málaga, Spain; 2Otorhinolaryngology Department, Clínica Universidad de Navarra, 31007 Pamplona, Spain; 3Otorhinolaryngology Department, Hospital Universitario de Fuenlabrada, Universidad Rey Juan Carlos, 28042 Madrid, Spain

**Keywords:** sleep apnea, myofunctional therapy, Iowa Oral Performance Instrument (IOPI), tongue strength, Tongue Digital Spoon

## Abstract

Myofunctional therapy (MT) is a recent treatment option for obstructive sleep apnea (OSA). The Iowa Oral Performance Instrument (IOPI) is a useful but expensive tool for measuring tongue strength in patients with OSA. We validated the Tongue Digital Spoon (TDS) to monitor tongue hypotonia in patients with OSA. Measurements with the IOPI and TDS were compared in patients with OSA before and after MT for tongue hypotonia. Baseline mean tongue strength measured with the IOPI in patients with moderate and severe OSA were 35.36 ± 9.05 and 33.83 ± 12.05, respectively, and that with the TDS were 168.55 ± 42.8 and 129.61 ± 53.7, respectively. After MT, mean tongue strength significantly improved: measured with the IOPI in patients with moderate and severe OSA were 53.85 ± 10.09 and 55.50 ± 9.64 (*p* = 0.8), and that with the TDS were 402.36 ± 52.92 and 380.28 ± 100.75 (*p* = 0.01), respectively. The correlation between the IOPI and TDS was high (r = 0.74; *p* = 0.01 pre-treatment, and r = 0.25; *p* = 0.05 post-treatment). The TDS is a useful tool for monitoring the efficacy of MT in patients with short-term OSA. Future randomized studies will determine the effectiveness of MT for the treatment of OSA.

## 1. Introduction

Obstructive sleep apnea (OSA) is a multifactorial disorder. Recently identified pathophysiological contributing factors include collapsibility of the airways, poor responsiveness of the pharyngeal muscles, a low excitation threshold, and increased loop gain [1,2]. Thus, understanding pathophysiology of every case is vital for selecting the most effective treatment option. It is well documented that conventional treatments (continuous positive airway pressure (CPAP), upper airway surgery, and oral appliances) may not always be successful in the presence of nonanatomical traits, especially in mild to moderate OSA.

Myofunctional therapy (MT) consists of isotonic and isometric exercises aimed at the oral and oropharyngeal structures, with the purpose of increasing muscle strength, tone, endurance, and coordinated movements of the pharyngeal and peri pharyngeal muscles. Recent studies have shown the effectiveness of MT in reducing snoring, apnea–hypopnea index (AHI), daytime sleepiness, and sleep quality [3,4]. MT also helps reposition the tongue, improves nasal breathing, and increases muscle tone in both pediatric patients and adults with OSA [5,6]. Indeed, studies have shown that MT prevents residual OSA in children after adenotonsillectomy and improves adherence in patients with OSA treated with CPAP [7].

The efficacy of MT in OSA patients is difficult to quantify, and long-term sleep studies (respiratory polygraphy and polysomnography (PSG)) are needed to monitor its results. Recent meta-analysis studies show that MT reduces the AHI by 34% in adults and improves the Epworth Sleepiness Scale (ESS) by 4 points and the lowest oxygen (O_2_) saturation by 2% [8]. The main drawback of MT is its low adherence, reported as being as high as 40% [4,5,6]. Thus, the incorporation of mobile apps to perform oropharyngeal exercises has improved the adherence percentages to MT and helped monitor the correct performance of these exercises [9].

However, measuring tongue strength may also be an excellent way to assess the efficacy of MT and monitor its role in OSA patients, as reported in patients with dysphagia [10]. Several authors have stated that the Iowa Oral Performance Instrument (IOPI) is a useful tool for measuring tongue strength in patients with OSA [4,11]. One disadvantage is that it is not affordable for all patients and doctors due to its economic limitations [4]. Therefore, the Tongue Digital Spoon (TDS) was also introduced as a valid instrument to monitor patients with tongue hypotonia and OSA [12,13].

The main objective of this study was to evaluate the strength of the genioglossus muscle with the TDS compared with the IOPI, the gold standard in patients with moderate and severe OSA before and after MT was applied, while CPAP was the primary treatment. 

## 2. Materials and Methods

A prospective nonrandomized pilot study was designed to include 51 patients with OSA divided into two groups (moderate N = 31 and severe N = 18). IOPI (model 2.1; IOPI Medical LLC, Carnation, WA, USA) [14] and TDS [12] were applied for the detection of patients with tongue hypotonia before treatment with MT following our recently published Tongue + protocol [13]. For the definition of OSA, we used PSG and subjective scales: ESS [15] and the Pittsburgh Sleep Quality Index (PSQI) [16]. Our speech therapy team conducted the sensory-motor evaluation [17]. We monitored the effectiveness of treatment using again IOPI, TDS, ESS, and PSQI after three months of MT. The oropharyngeal exercises were performed with the digital App Airway Gym^®^. The App has a total of 9 exercises to strengthen the oropharyngeal muscles based on proprioceptive training interacting with the screen of their smartphones by a sensible plugin. This app has the option to follow the accuracy and performance of the exercises by storing them in the cloud. This app offers acoustical, mechanical, and visual feedback to the patients and has proved its efficacy in previous studies [9]. Patients perform full sessions of 15 min a day for 5 days a week until completing 90 sessions.

### 2.1. Patients

The sample size was calculated from the results reported in previous studies using the TDS [12]. Based on an alpha level of 0.05 and a power of 0.80, it was estimated that 50 participants would be needed. The sample size was calculated using the program M (v.16, Addinsoft, Paris, France, XLSTAT), resulting in 50 patients. The participants gave written informed consent and the local Ethics Committee approved the study. The patients were between 18 and 75 years old and all received CPAP treatment after the diagnosis of PSG. All patients had a mean of 2–6 months of CPAP use.

### 2.2. Sleep Laboratory

Polysomnography was performed according to the technical specifications of the American Academy of Sleep Medicine [18]. The defining criteria for apnea and hypopnea with PSG were: hypopnea ≥30% decrease in airflow signal amplitude lasting ≥10 s and accompanied by ≥3% O_2_ desaturation. Apnea: a decrease of ≥90% in the amplitude of the airflow signal with a duration of ≥10 s. Absence of sleep apnea was defined as AHI < 5 events/h of sleep, moderate OSA with 15–29.9 events/h of sleep, and severe OSA with ≥30 events/h of sleep. 

### 2.3. Inclusion and Exclusion Criteria

Patients with moderate to severe OSA (AHI > 15), complete dentition, and grade 1–4 tonsils were included. The exclusion criteria are also collected in Table 1. Patients were excluded if they met one or more of the exclusion criteria. The most relevant of all, the presence of a muscle rehabilitation treatment, presence of ankyloglossia or some previous OSA treatment (surgery or mandibular advancement device), could affect the response to MT. We exclude patients with previous treatment of orofacial musculature in order to obtain the maximum answer to exercises obtained by this protocol. There might be patients that had rehabilitation in the past with remaining tongue strength values despite giving up the exercises [19].

### 2.4. IOPI

The IOPI was used to assess the strength of the lingual muscles. The reference values of the IOPI and how to use the device can be found in the reference manual [14,20]. The main measurements of the IOPI focus on evaluating tongue muscle strength and perioral muscle strength. It is performed by compressing a balloon connected to a device which the patient inserts into their mouth. To measure the maximum anterior tongue strength, the patient is asked to perform a peak compression of the balloon while resting on the papilla. This value corresponds to the strength of the genioglossus muscle. Three measurements are taken with a one-minute rest between each. The highest value obtained is taken as a reference. For the buccinator muscles, the balloon is placed between the gingival mucosa and the cheek, and the patient is asked to make a contraction with it. The value obtained corresponds to the tone of the buccinator muscle. The measurements are obtained in kilopascals (kPa). Tables of reference values obtained from the normal population allow patients to determine the condition of their muscles at the time of the test, and where they should ideally be according to their age and sex.

### 2.5. TDS

The TDS [12] is a kitchen utensil used to estimate food weight. To develop the TDS, we used the Soehnle Cooking Star digital measuring spoon with graduations from 0.1 to 500 g (ID ID20005876833). The TDS is a handheld instrument with a spoon that can be found on online shopping platforms. It has a handle with “tare” and “retention” buttons. Pressing the “hold” button gives the highest tare value, equivalent to IOPI maximum pressure. The spoon is inverted to perform the measurements, and a circular sticker of 1 cm^2^ is placed on the back, giving a clearly marked circumference. To measure tongue strength, the subject holds the spoon by the handle and, with their elbow resting on a flat surface, places the spoon on the tongue at an angle of approximately 30° from the elbow. The subject must calibrate the device by pressing the “hold” key, marking 0.0 g. The subject then presses the marked circumference with the tip of the tongue. Once this has been completed, the subject presses the “hold” button with the index finger holding the handle. This test is performed entirely by the subject to prevent movements in the spoon that may interfere with the results (Appendix A).

### 2.6. Statistical Analysis

IBM SPSS Statistics software 22.0 was used to analyze the study variables. To analyze the variables of IOPI pre/post-treatment, TDS pre/post-treatment, ESS pre/post-treatment, and PSQI pre/post-treatment between each of the groups of patients with moderate and severe OSA, the parametric T-test was used. A *p*-value of <0.05 was considered significant. Pearson’s correlation coefficient was used to evaluate the correlation between the two measuring instruments (IOPI vs. TDS). The examiners assessed the results prospectively and were unaware of the participants’ identities.

## 3. Results

Of the 35 patients with moderate OSA and 20 patients with severe OSA recruited for the study, 4 patients with moderate OSA and 2 with severe OSA dropped out because they stopped exercising and did not complete the 3 months of MT. In our study, we obtained an adherence rate of 89%. The final sample included 31 patients with moderate OSA and 18 with severe OSA.

### 3.1. Main Features

The patients included were middle-aged and overweight males with BMI > 25. As a condition of the study, none were obese (BMI > 30). All 49 patients with OSA used the CPAP as the first line of treatment. Initial evaluation of the BMI and the AHI are shown in Table 2.

### 3.2. Analysis of Variables

When we analyzed the variables in OSA patients before treatment with MT (Airway Gym^®^ app), significant differences in all the study variables were found as shown in Table 3 Once MT was applied, we performed the analysis between both groups (moderate versus severe apnea), and we found significant differences in all variables, including EES: 9.2 ± 2.4 vs. 10.3 ± 2.2 (*p* = 0,04), PSQI: 7.7 ± 2.5 vs. 12.4 ±2.3 (*p* = 0.01), IOPI tongue: 35.1 ± 14.8 vs. 54.7 ± 10.1 (*p* = 0.02), IOPI lips: 21.7 ± 9.7 vs. 31.5 ±8.6 (*p* = 0.03), and TDS 149.4 ± 46.8 vs. 397.3 ± 93.4 (*p* = 0.01) after MT (Table 4). IOPI and TDS are significantly correlated (Figure 1 and Figure 2).

## 4. Discussion

The main objective of our study was to compare two quantitative instruments for measuring muscular strength of the tongue, IOPI and TDS, as monitors of MT in OSA patients under CPAP treatment. We also intended to highlight TDS because it is easier to achieve at a reasonable cost. Finally, we demonstrated that monitoring OSA patients with IOPI and TDS were valid in both cases to observe the correct performance of orofacial exercises and promote their therapeutic adherence.

MT has been shown to provide a sustained increase in pharyngeal dilator tone, especially the genioglossus, in all stages of sleep [21]. It has also been used with conventional treatments, such as oral appliances, upper airway surgery, and CPAP, for moderate to severe OSA in the adult and pediatric populations. This suggests a need to improve muscle response capacity to obtain long-term results.

However, in a systematic review published in 2020, the absence of studies analyzing sleep quality in patients treated with MT and/or CPAP is notable [22]. Diaféria et al. [23] evaluated the final scores of the ESS and found that, compared with CPAP combined with MT, CPAP on its own may result in little or no difference in daytime sleepiness in 49 patients. In our study, comparing patients with moderate and severe apnea treated with CPAP, there were significant differences between the two groups after conducting MT (in moderate OSA, from 11 ± 3.53 to 9.2 ± 2.4; in severe OSA, from 15.8 ± 2.4 to 10.3 ± 2.2, *p* = 0.04).

Guimaraes et al. [24], in a study of 31 participants comparing the impact of MT and sham therapy on sleep quality, showed that MT could increase PSQI compared with sham therapy. Our results found significant differences in the final PSQI scores between the moderate and severe groups (7.7 ± 2.5 vs. 12.4 ± 2.3; *p* = 0.01).

In addition, the recent meta-analysis by Franciotti et al. [25] demonstrates the improvement in patients after experimental tongue training exercises, mainly when a worse value of maximum tongue pressure with the IOPI was found in the initial conditions. It concluded that IOPI proved to be a valid tool to successfully measure tongue pressure and detect the productive effects of tongue training in both healthy and diseased subjects. As we demonstrated in the previous article on the TDS [12], it is a tool validated with the IOPI and can be useful for home monitoring of the strength of the genioglossus musculature.

Guilleminault et al. [5] retrospectively analyzed the efficacy of postoperative MT, applied to 24 children (AHI: 0.4 ± 0.3) treated with adenotonsillectomy and palatal expansion. That study included 11 children who received MT after surgery, compared with 13 children who did not receive MT (controls). After monitoring for four years, the control group showed a recurrence of OSA (AHI: 5.3 ± 1.5/h), while the postoperative MT group remained cured without recurrence. This shows that removing the obstruction does not necessarily increase muscle responsiveness and underlines the role of MT in a multimodal approach. Hence, their study highlights the importance of monitoring the motor strength of the orofacial musculature. 

Villa et al. [6] observed a significant increase in objective measurements of tongue strength with the IOPI (31.9 ± 10.8 vs. 38.8 ± 8.3, *p* = 0.000), maximum tongue pressure (34.2 ± 10.2 vs. 38.1 ± 7.0, *p* = 0.000), and endurance (28.1 ± 8.9 vs. 33.1 ± 8.7, *p* = 0.01) in children with sleep-disordered breathing. We found similar results, this time comparing both IOPI and TDS data, observing a good correlation between them (see Table 4).

The published mean values of the TDS in the healthy population are 115.99 ± 170.47 g/cm^2^ in the youth group, 98.47 ± 110.51 g/cm^2^ in the middle-aged group, and 84.23 ± 140.53 g/cm^2^ in the elderly group [12]. In our series of patients, most of them were middle-aged men with moderate or severe sleep apnea. The mean TDS values were 118.5 ± 42.8 vs. 109 ± 53.7 g/cm^2^ (*p* = 0.23) before MT.

In this study, we observed that the range of values needed to observe an improvement in MT is very extensive due to the values of the spoon (0.1–500 g). With our results, we defined an improvement in the motor strength of patients using MT as having values greater than 350 g/cm^2^. Additionally, based on our three-year experience using this tool [26,27], patients with moderate to severe apnea with initial TDS values between 300 and 400 g/cm^2^ would not be candidates for MT exclusively as they have normal muscle strength.

In summary, we can highlight that we evaluated patients on CPAP plus MT and observed a clinical improvement in both IOPI and TDS values. However, this research has some limitations, mainly that we did not study the AHI after treatment. The number of patients was limited in both groups; the age range of 40–60 years was narrow and included a low number of women. 

In the latest published consensus on sleep apnea [28], MT is included as a therapeutic measure in non-obese patients with mild to moderate OSA, as a concomitant treatment to improve the efficacy and tolerance of CPAP by reducing the pressure required, and finally, as a complementary treatment in patients with an oral appliance. This work assumes the importance of monitoring MT with measuring instruments and questionnaires. For future research, we recommend expanding the topic: first, to investigate the effects of MT monotherapy in patients with moderate to severe apnea without determining anatomical factors, and, second, to investigate the patterns of muscle activation and response after MT through long-term PSG studies.

We recommend optimizing the results of the rehabilitation of the oropharyngeal muscles in patients with OSA, collaborating with speech therapists, and having instruments in consultation that monitor the increase in muscle strength, such as the IOPI and the TDS, obtained after a correct performance of the exercises.

## 5. Conclusions

The TDS is a valuable tool for monitoring the tongue motor strength of patients with moderate to severe sleep apnea treated with MT and CPAP.

## Figures and Tables

**Figure 1 life-12-01841-f001:**
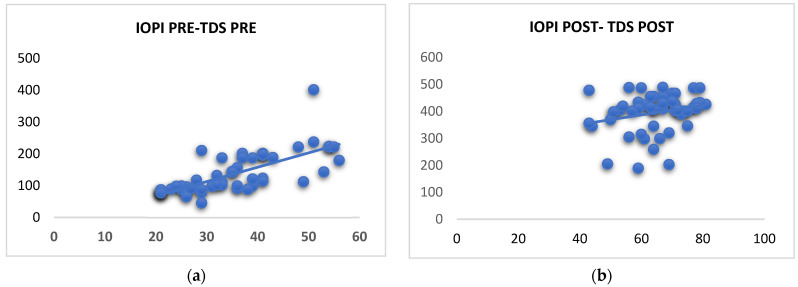
Correlation between the IOPI and the TDS in the measurement of tongue strength in patients with moderate to severe sleep apnea before oropharyngeal exercises was significantly positive. (**a**) *r* = 0.74; *p* = 0.01. After the treatment, the correlation was not significantly positive. (**b**) *r* = 0.25; *p* = 0.05.

**Figure 2 life-12-01841-f002:**
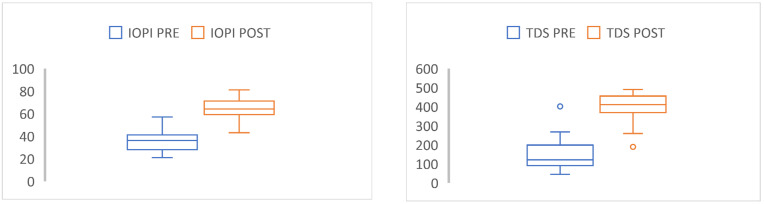
Box plot showing statistically significant changes before and after MT treatment with the Airway Gym^®^ app during 3 months in IOPI and TDS.

**Table 1 life-12-01841-t001:** Principal inclusion and exclusion criteria.

Inclusion Criteria	Exclusion Criteria
Moderate to severe OSA (AHI > 15)	BMI > 32 kg/m^2^
Grade I–IV tonsils	Ankyloglossia
Age between 18 and 75 years	Unstable coronary disease
Complete dentition	Hypnotic medication
	Systemic disease with a known inflammatory state
	Severe nasal obstruction
	Previous treatment for OSA or musculature rehabilitation

OSA, obstructive sleep apnea; BMI, body mass index.

**Table 2 life-12-01841-t002:** Characteristics of patients with mild and severe obstructive sleep apnea (OSA).

	Moderate OSA (N = 31)		Severe OSA (N = 18)	
Variable	Mean	SD	Mean	SD
Age (years)	52.7	6.6	51.1	10.2
BMI (kg/m^2^)	25.4	4.1	26.4	5.3
AHI (e/h)	23.9	6.2	41.2	8.3

*t*-test. *p* < 0.05 statistically significant difference. SD, standard deviation. e/h, event/hour. BMI, body mass index AHI, apnea–hypopnea index.

**Table 3 life-12-01841-t003:** Baseline data in the group of patients with moderate and severe OSA before treatment with the Airway Gym^®^ app.

	Moderate OSA (N = 31) (SD)	Severe OSA (N = 18) (SD)	*p* ^a^
Epworth Scale	11 (3.53)	15.8 (2.4)	0.41
Pittsburgh Scale	10.3 (2.1)	16.9 (2.5)	0.73
IOPI tongue (kPa)	35.6 (9.05)	32.8 (12.4)	0.08
IOPI lips (kPa)	23 (9.6)	22 (8.9)	0.45
TDS (g/cm^2^)	168.5 (42.8)	129.8 (53.7)	0.23

^a^ Student’s *t*-test. IOPI, Iowa Oral Performance Instrument; TDS, Tongue Digital Spoon. SD, standard deviation.

**Table 4 life-12-01841-t004:** Variables analyzed in all patients with OSA before and after MT treatment with the Airway Gym^®^ app during 3 months.

	Before Airway Gym^®^ Mean (SD) (N = 49)	After Airway Gym^®^ Mean (SD) (N = 49)	*p*
Epworth Scale	13.7 (4.2)	11.1 (2.6)	0.04
Pittsburgh Scale	15.2 (7.5)	9.8 (6.9)	0.02
IOPI tongue (kPa)	35.1 (14.8)	54.7 (10.1)	0.03
IOPI lips (kPa)	21.7 (9.7)	31.5 (8.6)	0.03
TDS (g/cm^2^)	149.4 (46.8)	397.3 (93.4)	0.01

IOPI, Iowa Oral Performance Instrument; TDS, Tongue Digital Spoon.

## Data Availability

Not applicable.

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
