# Peer review of "Evaluation of the Muscle Strength of the Tongue with the Tongue Digital Spoon (TDS) in Patients with Obstructive Sleep Apnea"

_life, 2022, doi:10.3390/life12111841_

Round 1

Reviewer 1 Report

Comments on ”Evaluation of the muscle strength of the tongue with the Tongue Digital Spoon (TDS) inpatients with obstructive sleep apnea”

An overall well-written manuscript. However:

1/ Abstract: 6th line: it is missing the word “severe”

                          There is no variety accompanying the values presented

2/ Materials and methods: 1st section: There is too little given about what exercises the training included and frequency of the training, or how the patients were instructed, followed and reinstructed if necessary.

3/ 2.1. last sentence: You write M, why not 50 if that is what you mean?

4/ You give many examinations performed which not corresponds to any results, either remove the examinations from the M&M or give some results of the examinations

5/ 2.3. What do you mean by “a history of rehabilitative treatment of the orofacial musculature”? Orofacial pain and dysfunction of the jaws in the population is high and many patients have had a training program, maybe many years ago. Are such patients excluded? And if so, why?

6/ Results, last section. You give a lot of figures already given in the tables. Please, remove the double information.

7/ The Figures and their legends miss information about the description/variety

8/ Discussion, 5th section: You write “a tool to measure tongue pressure” but you only measure the tongue-pressure in one direction - forwards. If you consider occlusal changes following tongue-pressure against or in between the front teeth, your training tool TDS has no use.

9/ Discussion, section 9, last sentence: This sentence is difficult to understand

Author Response

REVIEWER 1

1/ Abstract: 6th line: it is missing the word “severe”

Added in the text

2/ Materials and methods: 1st section: There is too little given about what exercises the training included and frequency of the training, or how the patients were instructed, followed and reinstructed if necessary.

The oropharyngeal exercises were performed with the digital App AirwayGym®. The App has a total of 9 exercises to strengthen the oropharyngeal muscles based on proprioceptive training interacting with the screen of their smartphones by a sensible plugin. This app has the option to follow the accuracy and performance of the exercises by storing them in the cloud. This app offers acoustical, mechanical, and visual feedback to the patients and has proved its efficacy  in previous studies [9]. Patients perform full sessions of 15 minutes a day for 5 days a week until 90 sessions.

3/ 2.1. last sentence: You write M, why not 50 if that is what you mean?

The sample size was calculated from the results reported in previous studies using the TDS [12]. Based on an alpha level of 0.05 and a power of 0.80, it was estimated that 50 participants would be needed. The sample size was calculated using the program M (v.16, Addinsoft, Paris, France, XLSTAT), resulting in 50 patients.

4/ You give many examinations performed which not corresponds to any results, either remove the examinations from the M&M or give some results of the examinations

These examinations are protocolized in all myofunctional studies in order to select the proper patient for this therapy. In this manuscript, we focused in the results obtained by IOPI and TDS.

5/ 2.3. What do you mean by “a history of rehabilitative treatment of the orofacial musculature”? Orofacial pain and dysfunction of the jaws in the population is high and many patients have had a training program, maybe many years ago. Are such patients excluded? And if so, why?

We wanted to exclude previous treatment with myofunctional or speech therapy of the orofacial muscles. We also excluded patients with severe pathology of the TMJ or malocclusions that difficult to perform the oral exercises.

We exclude patients with previous treatment of orofacial musculature in order to obtain the maximum answer to exercises obtained by this protocol. There might be patients that had rehabilitation in the past remained tongue strength values despite giving up the exercises.

We have added this quote: Oh JC. Effects of Tongue Strength Training and Detraining on Tongue Pressures in Healthy Adults. Dysphagia. 2015 Jun;30(3):315-20. doi: 10.1007/s00455-015-9601-x.

6/ Results, last section. You give a lot of figures already given in the tables. Please, remove the double information.  

We think it is important to have as many information as possible to help readers. Following your suggestion, Figure 2 has been simplified.

7/ The Figures and their legends miss information about the description/variety

They were added in the text:

Figure 2. Box plot showing statistically significant changes before and after MT treatment with the Airway Gym® app during 3 months in IOPI and TDS.

8/ Discussion, 5th section: You write “a tool to measure tongue pressure” but you only measure the tongue-pressure in one direction - forwards. If you consider occlusal changes following tongue-pressure against or in between the front teeth, your training tool TDS has no use.

Exactly, we are not talking about occlusal changes.

We simply see the tonicity that the tongue has when it comes to generating pressure.

This helps us to categorize patients as hypotonic and if you suspect a more important underlying disorder, they are referred to a specialist in speech therapy.

9/ Discussion, section 9, last sentence: This sentence is difficult to understand

Based on our three-year experience using this tool, patients with moderate-to-severe apnea with initial TDS values between 300 and 400 g/cm2 would not be candidates for MT exclusively as they have normal muscle strength.

Reviewer 2 Report

Thank you for your submission. The use of MT in the management of OSA could certainly be a useful component of an overall treatment plan. 

The lack of an AHI post treatment is somewhat disappointing as this would have been useful to further assess the utility of the technique.

You suggest using MT as mono therapy. This is unlikely to be feasible and would likely not be approved by most ethics boards as there is a risk of ongoing hypoxia if it is not successful, which can be effectively managed using established techniques.

Author Response

REVIEWER 2

Thank you very much for your comments. We will take them into account for next works, and effectively as we have carried out the study, the patients were being treatment with CPAP.